# Cutoff Values for Providing the Ideal Intravenous Patient-Controlled Analgesia According to the Intensity of Postoperative Pain—A Retrospective Observational Study

**DOI:** 10.3390/medicina57101065

**Published:** 2021-10-06

**Authors:** Keum Young So, Sang Hun Kim

**Affiliations:** 1Department of Anesthesiology and Pain Medicine, School of Medicine, Chosun University, 309 Pilmun-daero, Dong-Gu, Gwangju 61452, Korea; kyso@chosun.ac.kr; 2Department of Anesthesiology and Pain Medicine, Chosun University Hospital, 365 Pilmun-daero, Dong-Gu, Gwangju 61453, Korea

**Keywords:** fentanyl, non-opioid analgesics, opioid, patient-controlled analgesia, postoperative pain, rescue analgesics, rescue antiemetics

## Abstract

*Background and Objectives:* The cutoff values were analyzed for providing the ideal intravenous patient-controlled analgesia (PCA) that could reduce rescue analgesics or antiemetics requirements, based on the grades of postoperative pain intensity (PPI). *Materials and Methods:* PCA regimens of 4106 patients were retrospectively analyzed, and they were allocated into three groups with low, moderate, and high PPI grades (groups L, M, and H, respectively) based on numeric rating scores obtained 6 h postoperatively. Opioid and non-opioid analgesic doses were converted into fentanyl-equivalent doses (DOSE-FEN-OP and DOSE-FEN-NONOP, respectively). The primary endpoint was the cutoff values of these parameters. *Results:* With respect to the PCA settings to reduce rescue analgesic and antiemetic requirements, group L required a background infusion rate (BIR) of 1.75–3 mL/h, bolus volume of 0.5–1.25 mL, and lockout interval of ≤12.5 min. Group M required a BIR of 1.75 mL/h, bolus volume of 0.5–1.75 mL, and lockout interval of ≤5 min. Group H required a BIR of 1.75 mL/h, bolus volume of 0.5 mL, and lockout interval of ≤5 min. In assessments of the analgesic doses to reduce rescue analgesic requirement, the DOSE-FEN-OP was at least 950 μg of fentanyl regardless of group, while the DOSE-FEN-NONOP was ≥250 μg, ≥550 μg, and ≥700 μg for the L, M, and H groups, respectively. In assessments of the analgesic doses to reduce rescue antiemetic requirement, DOSE-FEN-OP was ≤950 μg for groups L and M and ≤850 μg for Group H, while DOSE-FEN-NONOP was ≤50 μg, ≤450 μg, and ≤700 μg for groups L, M, and H, respectively. *Conclusion:* The ideal PCA for reduction in rescue analgesics or antiemetics can be achieved by adjustment of PCA settings and drug dosages carefully with these cutoff values depending on the expected grades of PPI. Especially, the ideal PCA can be provided by adjusting the lockout interval and bolus volume rather than BIR and by applying smaller bolus doses and shorter lockout intervals with an increasing PPI grade.

## 1. Introduction

Intravenous patient-controlled analgesia (PCA) has become the most common standard modality for postoperative pain control worldwide and is associated with high satisfaction rates [1,2]. Combinations of opioids, non-opioid analgesics, and antiemetics are usually adopted for intravenous PCA, considering the need for reduced opioid doses, the opioid-sparing effects of non-opioid analgesics, and the reduced incidence of postoperative nausea and vomiting (PONV) [3]. However, because of the lack of a consensus on the appropriate dose of opioids and adjuvants [1,2], opioid-based PCA is associated with the risk of PONV or insufficient analgesia if the opioid doses are inappropriate. In this situation, patients commonly require rescue analgesics or antiemetics for controlling these adverse events.

Fentanyl has been popularly adopted as a more appropriate and suitable opioid than morphine for intravenous PCA due to its more rapid onset and shorter duration of action, fewer opioid-related adverse events, and higher satisfaction score than morphine [1,4,5]. However, textbook-recommended fentanyl doses for intravenous PCA are still not widely used because these fentanyl doses are thought to be slightly excessive for use in Korean patients. Furthermore, the PCA regimens are usually determined by the attending anesthesiologist on the basis of their preference and judgment, and various PCA device settings (background infusion rate (BIR), bolus volume, and lockout interval), and various doses of fentanyl with or without adjuvant analgesics and antiemetics are used in these regimens. However, the provision of optimal postoperative analgesia without adverse events remains difficult because of inadequate pain control due to patient-level differences in postoperative pain intensities, individual opioid requirements, and unadjustable risk factors [2,6]. Thus, several patients receiving PCA may require rescue analgesics due to inadequate postoperative analgesia and rescue antiemetics or discontinuation of PCA due to opioid-related adverse events.

Considering this situation, the development of recommendable cutoff values for PCA settings and drug dosages to provide the ideal intravenous PCA based on clinical situations is essential. However, the evidence for proper fentanyl use in PCA is limited since most studies were conducted with morphine-based regimens [1,7]. Therefore, a retrospective review of electronic medical records was performed to investigate the cutoff values for PCA settings and drug dosages to provide the ideal intravenous fentanyl-based intravenous PCA that can reduce rescue analgesic and antiemetic requirements on the basis of postoperative pain intensity (PPI) regardless of surgical department and surgical type.

## 2. Materials and Methods

### 2.1. Study Design and Ethics Statement

The institutional review board (IRB) of Chosun University Hospital approved this retrospective study by electronic medical record review (approval number: CHOSUN 2018-12-008) on 3 January 2019. The IRB also waived the need to obtain written informed consent from patients because the patients’ identifying information was anonymized before the analysis, and this study did not pose more than minimal risk to subjects. This study was prospectively registered with the Clinical Research Information Service (CRIS: https://cris.nih.go.kr/, ref: KCT0003889) on 7 May 2019 and was conducted according to the Declaration of Helsinki of 1964 and all its subsequent revisions.

### 2.2. Selection of the Study Population

This study enrolled 4151 patients aged 12–100 years who received intravenous PCA, had an American Society of Anesthesiologists physical status (ASA PS) of I–III and were scheduled to undergo any elective surgery from 3 January 2019 to 29 December 2020. Patients with cognitive disorders (*n* = 30), unstable hemodynamics requiring treatment in intensive care units (*n* = 15), and a history of receiving any type of nerve block or skin infiltration of local anesthetics additionally (*n* = 0) were excluded from this study. Finally, 4106 patients were enrolled in this study (Figure 1).

### 2.3. Anesthetic Management

After premedication with intramuscular midazolam or no premedication, the patients were transferred to an operating room. All patients received either general anesthesia (inhaled or balanced anesthesia), total intravenous anesthesia, or regional anesthesia. A 50% oxygen-air or medical air mixture was used during mechanical ventilation in patients receiving general anesthesia. Consistent hypotension was controlled with an intermittent bolus volume of either 100 µg phenylephrine or 10 mg ephedrine, while consistent high blood pressure was controlled with an intermittent bolus volume of 1 mg nicardipine. Bradycardia below 50 beats/min was controlled with an intermittent bolus volume of 0.5 mg atropine. Tachycardia above 120 beats/min was controlled with an intermittent bolus volume of 10 mg esmolol. Intraoperative hypothermia was prevented with the application of an air-forced blanket warmer. Appropriate neuromuscular blockers for neuromuscular paralysis were used based on the patient’s underlying diseases, and their effects were fully reversed by administration of sugammadex, glycopyrrolate, and pyridostigmine, or all three agents. In patients receiving intraoperative opioids, persistent opioid-related respiratory nonresponse was stimulated with 0.1 mg naloxone intermittent injection during emergence. Persistent sedation with midazolam premedication was reversed with 0.3 mg flumazenil during emergence.

### 2.4. Interventions

Every application of PCA was performed in accordance with the hospital protocol for postoperative pain management. On the day before surgery, anesthesiologists explained the usage of the PCA devices to all patients, who agreed to use intravenous PCA for postoperative analgesia. For PCA devices with bolus dosing, the patients were instructed to push the “demand” button of each device whenever they experienced pain of >4 points on the numeric rating scale (numerical rating scale (NRS): 0 = no pain, 10 = worst pain).

The attending anesthesiologists operated each PCA device at the end of the surgery. A total PCA volume of 100 mL, consisting of normal saline, opioids (fentanyl, sufentanil, or oxycodone), adjuvant analgesics (none, nefopam, or ketorolac), and adjuvant antiemetics (none or ramosetron), was used. Basically, all PCA devices were set with a BIR of 2 mL/h, bolus volume of 2 mL, and lockout interval of 30 min. However, the attending anesthesiologist had the choice to determine the decided drug dosage and device settings for PCA according to their preference and judgment, considering the patient’s safety.

In patients receiving PCA, rescue analgesics and antiemetics were administrated only on demand and not routinely. When patients experienced pain with an NRS score > 4, the patient pushed the “demand” button for the administration of a preset bolus volume. When patients required additional rescue analgesics within the lockout interval, physicians or nurses injected opioids, nonsteroidal anti-inflammatory drugs, or other analgesics. PONV (NRS > 4) was controlled by intravenous injection of 10 mg metoclopramide or 0.3 mg ramosetron.

The nurses, who were trained in the hospital to assess patients using the NRS, recorded the scores for postoperative pain and PONV, the rescue analgesics and antiemetics administered, and any adverse events in electronic medical records. Decisions to stop PCA were made by the anesthesiologists on the basis of the severity of patients’ signs and symptoms.

### 2.5. Outcomes

PCA regimens (types and doses of opioids, adjuvant analgesics, and adjuvant antiemetics) and PCA device settings (BIR, bolus volume, and lockout interval) were analyzed. Doses of opioids, non-opioid analgesics, and total analgesics were converted to fentanyl-equivalent doses (in μg; DOSE-FEN-OP, DOSE-FEN-NONOP, and DOSE-FEN-TOTAL, respectively) considering the ratios of oxycodone (μg) to fentanyl (100:1), sufentanil (μg) to fentanyl (1:10), ketorolac (mg) to fentanyl (25:100), and nefopam (mg) to fentanyl (20:100) [8,9,10,11]. Then, the BIRs were recalculated on the basis of these converted doses (BIR-FEN-OP, BIR-FEN-NONOP, and BIR-FEN-TOTAL). DOSE-FEN-TOTAL was the total fentanyl-equivalent analgesic dose converted from opioid and non-opioid analgesics.

The NRS at the sixth postoperative hour was analyzed to allocate patients into low, moderate, and high PPI groups (group L, group M, and group H, respectively) according to NRS > 4, 4 ≤ NRS < 7, NRS ≥ 7 [12]. Meanwhile, the use of rescue analgesics and antiemetics was analyzed along the 48th postoperative hour.

Age, sex, weight, body mass index (BMI), ASA PS, surgery department, PPI grade at the sixth postoperative hour, history of previous opioid intake, underlying diseases (diabetes mellitus, hypertension, chronic obstructive pulmonary disease, coronary disease, etc.), PONV risk factors (smoking, motion sickness, and previous PONV), anesthesia duration, and intraoperative opioid were also analyzed.

### 2.6. Analysis

The primary endpoints were the cutoff values of PCA settings, DOSE-FEN-OP, DOSE-FEN-NONOP, BIR-FEN-OP, and BIR-FEN-NONOP that could increase or decrease the probability of requiring rescue analgesic or rescue antiemetics.

All statistical analyses were performed with SPSS Statistics for Windows, ver. 26.0 (IBM Corp., Armonk, NY, USA). All data were presented as means (95% confidence intervals (CI)) or numbers (percentage) of patients (*n* (%)).

Receiver operating characteristic (ROC) curve analysis was performed to obtain cutoff values for PCA settings (BIR, bolus volume, and lockout interval), DOSE-FEN-OP, DOSE-FEN-NONOP, DOSE-FEN-TOTAL, DOSE-EME (antiemetics dose), BIR-FEN-OP, BIR-FEN-NONOP, BIR-FEN-TOTAL, and BIR-EME (background infusion rate of antiemetics) indicating the need for rescue analgesics or antiemetics. Optimal cutoff values were determined based on the maximum values of the Youden index, calculated by (sensitivity + specificity − 1). Statistical significance was set at *p* < 0.05.

Continuous variables were analyzed using the one-way analysis of variance (ANOVA) test, following Scheffe’s post-hoc test, while nominal variables were analyzed with the χ^2^ test or Fisher’s exact test. Statistical significance was set at *p* < 0.05.

## 3. Results

### 3.1. Characteristics of the Patients

In this study, 4106 patients were eligible for analysis (Figure 1). The patients’ characteristics are shown in Table 1. The three groups showed no significant differences in sex, age, height, weight, BMI, ASA PS, and anesthesia duration, and in the prevalence of underlying diseases, smoking, opioid-naïve status, and intraoperative opioid use (Table 1).

### 3.2. Drugs Used in Intravenous PCA

There were no significant intergroup differences in the opioids, adjuvant analgesics, and adjuvant antiemetics included in PCA regimens (Table 2).

### 3.3. Settings and Drug Doses in Intravenous PCA

Among the PCA settings, bolus volume and lockout interval showed significant differences among the three groups (*p* = 0.001 and *p* = 0.001, respectively) (Table 3). Bolus volume and lockout interval of group M were higher than that of group L (*p* = 0.002 and *p* = 0.002), but there were no significant differences between groups M and H.

DOSE-EME showed significant differences among the three groups (*p* < 0.001), while there were no significant intergroup differences in DOSE-FEN-OP, DOSE-FEN-NONOP, and DOSE-FEN-TOTAL (Table 3). DOSE-EME of group M was also higher than those of groups L and H (*p* = 0.002 and *p* = 0.002, respectively).

### 3.4. Background Infusion Rate of Opioids, Non-Opioid Analgesics, and Antiemetics for PCA

There were no significant intergroup differences in BIR-FEN-OP, BIR-FEN-NONOP, and BIR-FEN-TOTAL. BIR-EME was significantly different among groups (*p* = 0.014) and was lower in group H than in group L (*p* = 0.020, Table 4).

### 3.5. Requirement for Rescue Analgesics and Antiemetics during Intravenous PCA

The requirement for rescue analgesics was significantly different among the three groups (*p* < 0.001) and was the highest in group H (26.6%), followed by group M (20%) and group L (16.6%, Table 5). On the other hand, rescue antiemetic requirement was not significantly different among the three groups.

### 3.6. Cutoff Values of Potential Variables for Requiring Rescue Analgesics and Antiemetics

#### 3.6.1. Cutoff Values of Potential Variables for Requiring Rescue Analgesics

In patients with low PPI, the cutoff values for BIR, bolus volume, and lockout interval were 1.75 mL/h (Area under the ROC Curve (AUC): 0.515), 0.5 mL (AUC: 0.610), and 12.5 min (AUC: 0.619), respectively (Figure 2A and Appendix A). The cutoff values for DOSE-FEN-OP, DOSE-FEN-NONOP, and DOSE-FEN-TOTAL were 950 μg (AUC: 0.559), 250 μg (AUC: 0.501), and 1750 μg (AUC: 0.541), respectively (Figure 3A and Appendix A). For BIR-FEN-OP, BIR-FEN-NONOP, BIR-EME, and BIR-FEN-TOTAL, the cutoff values were 19 μg/h (AUC: 0.567), 7 μg/h (AUC: 0.509), 15 μg/h (AUC: 0.510), and 35 μg/h (AUC: 0.548), respectively (Figure 3C and Appendix A). The cutoff values for PCA settings (bolus volume (*p* = 0.001) and lockout time (<0.001)), DOSE-FEN-OP (*p* = 0.032), and BIR-FEN-OP (*p* = 0.01) showed statistical significance.

In patients with moderate PPI, the cutoff values for BIR, bolus volume, and lockout interval were 1.75 mL/h (AUC: 0.504), 0.5 mL (AUC: 0.524), and 5 min (AUC: 0.512), respectively (Figure 2A and Appendix A). The cutoff values for DOSE-FEN-OP, DOSE-FEN-NONOP, and DOSE-FEN-TOTAL were 950 μg (AUC: 0.612), 550 μg (AUC: 0.500), and 1750 μg (AUC: 0.583), respectively (Figure 3A and Appendix A). For BIR-FEN-OP, BIR-FEN-NONOP, BIR-EME, and BIR-FEN-TOTAL, the cutoff values were 19 μg/h (AUC: 0.610), 11 μg/h (AUC: 0.500), 21 μg/h (AUC: 0.504), and 35 μg/h (AUC: 0.581), respectively (Figure 3C and Appendix A). The cutoff values for PCA settings (bolus volume and lockout time) were not statistically significant, while those for DOSE-FEN-OP (*p* < 0.001), DOSE-FEN-TOTAL (*p* < 0.001), BIR-FEN-OP (*p* < 0.001), and BIR-FEN-TOTAL (*p* < 0.001) were statistically significant.

In patients with high PPI, the cutoff values for PCA settings (BIR, bolus volume, and lockout interval) were 1.75 mL/h or lower (AUC: 0.508), 0.5 mL (AUC: 0.573), and 5 min (AUC: 0.605), respectively (Figure 2Aand Appendix A). DOSE-FEN-OP, DOSE-FEN-NONOP, and DOSE-FEN-TOTAL were 950 μg (AUC: 0.660), 700 μg (AUC: 0.540), and 1550 μg (AUC: 0.656), respectively (Figure 3A and Appendix A). For BIR-FEN-OP, BIR-FEN-NONOP, BIR-EME, and BIR-FEN-TOTAL, the cutoff values were 19 μg/h (AUC: 0.662), 14 μg/h (AUC: 0.546), 21 μg/h (AUC: 0.545), and 35 μg/h (AUC: 0.658), respectively (Figure 3C and Appendix A). The cutoff values for PCA settings (bolus volume (*p* = 0.002) and lockout time (*p* < 0.001)), DOSE-FEN-OP (*p* < 0.001), DOSE-FEN-TOTAL (*p* < 0.001), BIR-FEN-OP (*p* < 0.001), and BIR-FEN-TOTAL (*p* < 0.001) showed statistical significance.

#### 3.6.2. Cutoff Values of Potential Variables for Requiring Rescue Antiemetics

In patients with low PPI, the cutoff values for BIR, bolus volume, and lockout interval were 3 mL/h (AUC: 0.494), 1.25 mL (AUC: 0.576), and 17.5 min (AUC: 0.583), respectively (Figure 2B and Appendix A). The cutoff values for DOSE-FEN-OP, DOSE-FEN-NONOP, and DOSE-FEN-TOTAL were 950 μg (AUC: 0.615), 50 μg (AUC: 0.470), and 1350 μg (AUC: 0.543), respectively (Figure 3B and Appendix A). The cutoff values for BIR-FEN-OP, BIR-FEN-NONOP, BIR-EME, and BIR-FEN-TOTAL were 19 μg/h (AUC: 0.613), 1 μg/h (AUC: 0.468), 25 μg/h (AUC: 0.474), and 27 μg/h (AUC: 0.541), respectively (Figure 3D and Appendix A). However, the cutoff values for none of the potential variables showed statistical significance.

In patients with moderate PPI, the cutoff values for BIR, bolus volume, and lockout interval were 1.75 mL/h (AUC: 0.523), 1.75 mL (AUC: 0.519), and 25 min (AUC: 0.525), respectively (Figure 2B and Appendix A). The cutoff values for DOSE-FEN-OP, DOSE-FEN-NONOP, and DOSE-FEN-TOTAL were 950 μg (AUC: 0.627), 450 μg (AUC: 0.548), and 1550 μg (AUC: 0.619), respectively (Figure 3B and Appendix A). For BIR-FEN-OP, BIR-FEN-NONOP, BIR-EME, and BIR-FEN-TOTAL, the cutoff values were 19 μg/h (AUC: 0.634), 8.5 μg/h (AUC: 0.557), 21 μg/h (AUC: 0.532), and 31 μg/h (AUC: 0.626), respectively (Figure 3D and Appendix A). The cutoff values for PCA settings (bolus volume and lockout time) were not statistically significant, while those for DOSE-FEN-OP (*p* < 0.001), DOSE-FEN-TOTAL (*p* = 0.001), BIR-FEN-OP (*p* < 0.001), and BIR-FEN-TOTAL (*p* < 0.001) were statistically significant.

In patients with high PPI, the cutoff values for BIR, bolus volume, and lockout interval were 1.75 mL/h (AUC: 0.541), 0.5 mL (AUC: 0.491), and 12.5 min (AUC: 0.522), respectively (Figure 2B and Appendix A). The cutoff values for DOSE-FEN-OP, DOSE-FEN-NONOP, and DOSE-FEN-TOTAL were 850 μg (AUC: 0.614), 700 μg (AUC: 0.629), and 1450 μg (AUC: 0.670), respectively (Figure 3B and Appendix A). For BIR-FEN-OP, BIR-FEN-NONOP, BIR-EME, and BIR-FEN-TOTAL, the cutoff values were 17 μg/h (AUC: 0.641), 14 μg/h (AUC: 0.660), 21 μg/h (AUC: 0.539), and 29 μg/h (AUC: 0.700), respectively (Figure 3D and Appendix A). The cutoff values for PCA settings (bolus volume and lockout time) were not statistically significant, while those for DOSE-FEN-NONOP (*p* = 0.033), DOSE-FEN-TOTAL (*p* = 0.010), BIR-FEN-OP (*p* = 0.039), BIR-FEN-NONOP (*p* = 0.008) and BIR-FEN-TOTAL (*p* = 0.002) were statistically significant.

## 4. Discussion

This study identified the cutoff values of the settings and drug compositions for the ideal PCA regimen according to the grades of PPI. In general, higher bolus volume, faster BIR, shorter lockout interval, and larger opioid dose in the PCA settings were related to decreased demand for rescue analgesics, while they were risk factors for rescue antiemetic requirement. A previous study analyzed the cutoff values indicating no requirement of rescue analgesics and antiemetics in patients receiving fentanyl-based postoperative PCA [2]. They suggested that a fentanyl BIR should be at least 0.38 μg/kg/h to provide effective postoperative analgesia without administration of rescue analgesics and a fentanyl BIR of over 0.36 μg/kg/h to administer rescue antiemetic. Although the findings of that study included only the cutoff values for BIRs in general and in situations with or without the addition of adjuvant analgesics and antiemetics, they did not include more detailed cutoff values for PCA settings, drug doses, and the individual BIRs of opioids, non-opioid analgesics, and antiemetics. To provide effective postoperative analgesia, it is important to provide the ideal PCA regimen on the basis of the predicted PPIs of each patient. Thus, this study is meaningful because it analyzed the cutoff values of PCA parameters that would indicate no requirement of rescue analgesics and antiemetics in patients receiving postoperative PCA.

### 4.1. PCA Settings

As a basic concept, to reduce the requirement for rescue analgesia, the PCA device should be set with BIR and bolus volume values greater than the respective cutoff values and a lockout interval less than the cutoff value. On the other hand, to reduce the requirement for rescue antiemetics, the PCA device should be set with values less than the cutoff BIR, bolus volume, and lockout interval values. A shorter lockout interval may increase the administration of opioids, which could increase the risk of opioid-induced adverse effects. However, in cases involving a PCA regimen with premixed antiemetics as an adjuvant, it can also provide a counteracting effect that offsets side effects by increasing their administrated dosage. Thus, setting the lockout interval below the cutoff value can reduce the demand for rescue antiemetics.

The PCA setting should be changed according to the expected grades of PPI, considering effective postoperative analgesia without requiring rescue analgesics and antiemetics in patients receiving PCA premixed with analgesics and antiemetics.

For patients with a low expected PPI, this study showed that a BIR of 1.75–3 mL/h, bolus volume of 0.5–1.25 mL, and lockout interval of ≤12.5 min were required for effective analgesia without the need for rescue analgesics and antiemetics (Figure 2). For patients with moderate expected PPI, a BIR of 1.75 mL/h, bolus volume of 0.5–1.75 mL, and lockout interval of ≤5 min were required for effective analgesia without side effects (Figure 2). For patients with a high expected PPI, this study showed that a BIR of 1.75 mL/h, bolus volume of 0.5 mL, and lockout interval of ≤5 min were required for effective analgesia without side effects (Figure 2). These findings suggest that effective PCA can be provided by adjusting the lockout interval and bolus volume rather than BIR and by applying smaller bolus doses and shorter lockout intervals with an increasing PPI grade.

### 4.2. Doses and BIRs of Analgesics and Antiemetics

The doses and BIRs of opioids and adjuvants (non-opioid analgesics and antiemetics) also should be adjusted according to PPI grades because controlling PCA settings alone is not enough to achieve sufficient analgesia without adverse effects. A higher BIR of fentanyl is a double-edged sword that could decrease the demand for rescue analgesics and increase the demand for rescue antiemetics [2]. Shin et al. [2] also identified a lower BIR of fentanyl as a risk factor for the use of rescue analgesics and a higher BIR of fentanyl as a risk factor for the use of rescue antiemetics.

This study showed that, for a reduction in the demand for rescue analgesics, the fentanyl-equivalent opioid dose required was at least 950 μg (19 μg/h) regardless of PPI (Figure 3A,C). In addition, the fentanyl-equivalent non-opioid doses required were ≥250 μg (7 μg/h), ≥550 μg (11 μg/h), and ≥700 μg (14 μg/h) for patients with low, moderate, and high expected PPI, respectively (Figure 3A,C). On the other hand, for a reduction in the demand for rescue antiemetics, the fentanyl-equivalent opioid dose required was less than 950 μg (19 μg/h) in patients with low and moderate expected PPI and less than 850 μg (17 μg/h) in patients with high expected PPI (Figure 3C,D). In addition, the fentanyl-equivalent non-opioid dose was increased and was ≤50 μg (1 μg/h), ≤450 μg (8.5 μg/h), and ≤700 μg (14 μg/h) for patients with low, moderate, and high expected PPI, respectively (Figure 3C,D), which were less than those for reduction in the demand for rescue analgesics.

These findings suggest that there is no optimal dose and BIR of analgesics for reducing the demand for rescue analgesics and antiemetics and that it should be considered that if the dose or BIR of PCA drugs is set between these cutoff values to reduce the demand for rescue analgesics and antiemetics, the patients may experience uncontrolled postoperative pain, PONV, or both.

The cutoff values for the BIR of antiemetics were higher for reducing the demand for rescue analgesics depending on increasing expected grade of PPI, and they were higher in patients with low expected PPI but similar in patients with moderate and high expected PPIs (Figure 3c). However, to reduce the demand for rescue antiemetics, the cutoff value for the BIR of antiemetics was higher (25 μg/h) than that (15 μg/h) for reducing the demand for rescue analgesics in patients with low expected PPI but were similar (21 μg/h) in patients with moderate and high PPIs (Figure 3C,D). Thus, it can be set the BIR of antiemetics between 15 and 25 μg/h considering the risk–benefit ratio between effective analgesia and less adverse events.

Thus, it is necessary to adjust the PCA settings and doses of analgesics to provide effective analgesia without adverse events, and PCA should be modified to provide effective analgesia or to minimize opioid-induced adverse events, as appropriate.

### 4.3. Limitations of This Study

This study had some limitations. First, the AUCs of the cutoff values were relatively low due to the uneven distribution and low incidence of rescue analgesic and antiemetic usage. A randomized controlled trial using data with normal distribution is necessary to support these results.

Second, patients receiving various opioids and non-opioid analgesics during PCA were enrolled, and all analgesics were converted into fentanyl-equivalent doses using conversion ratios reported in previous studies [8,9,10,11]. Even though the conversion ratios between opioids are well known and validated, the conversion ratios between opioids and non-opioid analgesics are not.

Third, there were several variables at intraoperative, emergence, and postoperative periods, which could influence the postoperative PPI. Intraoperative opioid consumption during the intraoperative period is considered as one of the indicators that distinct surgical noxious stimulation or classify the grade of PPI according to surgeries. Some of the patients enrolled in this study received naloxone for persistent opioid-related respiratory nonresponse and flumazenil for persistent sedation during the emergence period, in accordance with hospital protocol for perioperative anesthesia management. The postoperative use of the demand button for infusion of PCA opioids, rescue analgesics, and rescue antiemetics also could influence the postoperative PPI. Unfortunately, the authors could not register these data because they were not recorded in electronic medical records completely according to a specified time interval during the perioperative period. Therefore, it was difficult to classify the grade of PPI by the surgery types or surgical departments in this retrospective study.

Forth, this retrospective study included many surgeries with various PPIs and rates of PONV, but subgroup analysis based on the types of surgery was not performed. Furthermore, the authors did not use the anticipated surgery-specified pain intensity for the classification of PPI grades. Actually, even if patients undergo the same surgery from a doctor, they can show varieties of PPI. Therefore, it is very difficult to determine the PPI specifically for each type of surgery [13,14]. In particular, it was very difficult to control risk factors to influence the PPI grade for each surgery because of the incompleteness of the data as the limitation of the retrospective study. However, the data of demography and PCA regimens were shown a minimized bias with no significant difference after classification of data into three groups based on PPIs at postoperative 6 h. Therefore, the authors used the PPI grades determined with the actual numeric rate score at the sixth postoperative hour in patients who received PCA. To validate these results, a well-designed randomized controlled trial or a retrospective study is necessary to confirm the effective procedure-specific regimens in the future.

Therefore, careful interpretation of the findings of this study is necessary to provide fentanyl-based PCA for effective postoperative analgesia in specified surgeries with different expected PPI, and further research will be required with the dosages and settings presented in this study.

Finally, this study was performed on Korean patients. Therefore, caution should be used when extrapolating the results of this study to the general population.

## 5. Conclusions

For the optimal or ideal regimens of PCA depending on PPI, the authors suggest that the adjustment for PCA settings is needed based on a BIR of 1.75 mL/h and bolus volume of 0.5 mL regardless of expected PPI and the lockout interval among PCA setting is needed to adjust within 12.5 min for cases with a low expected PPI, and within 5 min for those with a moderate or high expected PPI. Therefore, adjustment of the lockout interval should be considered more than those of BIR and bolus volume for the PCA setting.

For the optimal or ideal PCA regimens, drug combinations should also be considered depending on the degree of PPI. Basically, while maintaining 950 μg of fentanyl, increasing the dosage of non-opioid analgesics (with doses of fentanyl equivalent) could provide effective PCA, considering the expected increase in PPI.

However, as the degree of PPI increased, the cutoff values of some parameters did not overlap with the probability of requiring rescue analgesics or rescue antiemetics. This suggests that patients receiving PCA with settings and drug doses between the cutoff values for rescue antiemetics and those for rescue analgesics may suffer from uncontrolled postoperative pain or PONV, which is the worst-case scenario [2]. Thus, it is necessary to choose between minimizing the possibility of a rescue analgesic requirement or minimizing the possibility of a rescue antiemetic requirement. On the basis of this decision, the PCA setting, and drug dosage should be determined carefully.

Finally, the optimal fentanyl-based PCA could be provided by determining the setting and drug dosage of PCA, considering the cutoff values and risk/benefit factors calculated according to the expected degree of PPI. In addition, further research will be required to identify optimal regimens that can maximize PCA analgesic effects and minimize adverse events such as PONV.

## Figures and Tables

**Figure 1 medicina-57-01065-f001:**
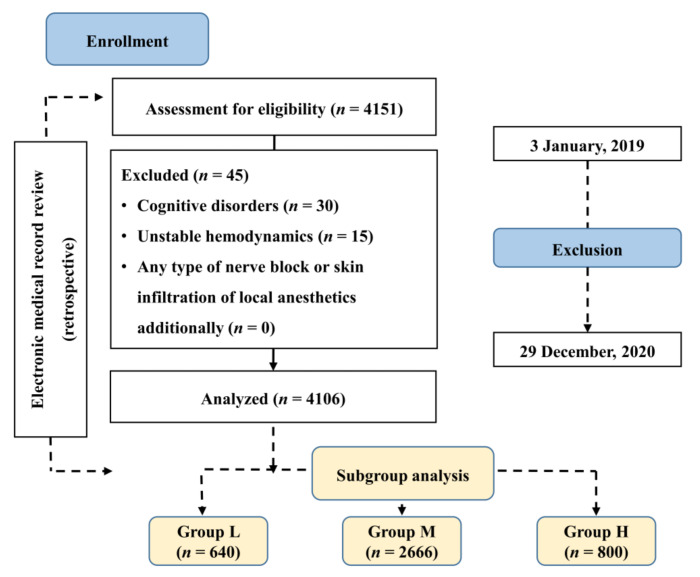
Flowchart of this study. Group L (*n* = 640), NRS < 4; Group M (*n* = 2666), 4 ≤ NRS < 7; Group H (*n* = 800), NRS ≥ 7 at the 6th postoperative hour.

**Figure 2 medicina-57-01065-f002:**
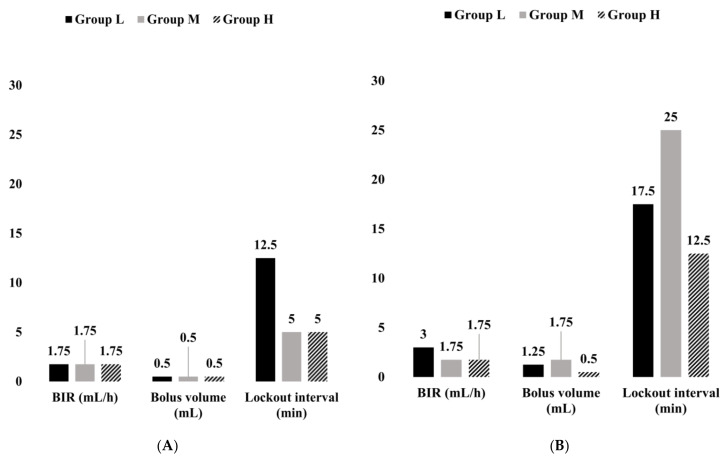
Cutoff values of PCA settings for reduction in rescue analgesic requirement (**A**) and rescue antiemetics (**B**) according to PPI. PPI; postoperative pain intensity. Group L (*n* = 640), NRS < 4; Group M (*n* = 2666), 4 ≤ NRS < 7; Group H (*n* = 800), NRS ≥ 7 at the 6th postoperative hour.

**Figure 3 medicina-57-01065-f003:**
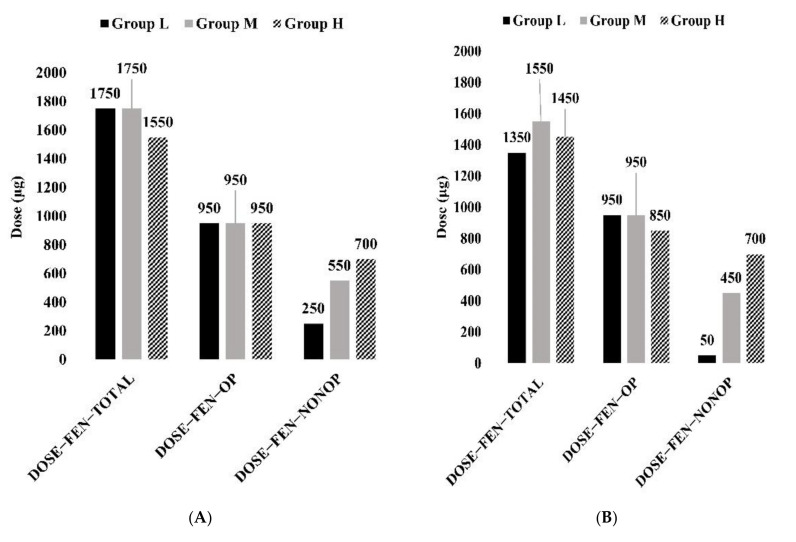
Cutoff values of doses (**A**,**B**) and BIR (**C**,**D**) for the reduction in rescue analgesic requirement (**A**,**C**) and rescue antiemetics (**B**,**D**) according to PPI. BIR, background infusion rate, PPI; postoperative pain intensity. Group L (*n* = 640), NRS < 4; Group M (*n* = 2666), 4 ≤ NRS < 7; Group H (*n* = 800), NRS ≥ 7 at the 6th postoperative hour.

**Table 1 medicina-57-01065-t001:** Characteristics of patients (*n* = 4106).

Variables	Group L (*n* = 640)	Group M (*n* = 2666)	Group H (*n* = 800)	*p*-Value
Female sex	317 (49.5)	1354 (50.8)	412 (51.5)	0.755
Age (years)	58.3 (56.9, 59.7)	57.3 (56.7, 58.0)	57 (55.8, 58.2)	0.367
Height (cm)	163.3 (162.6, 164.1)	163.4 (163, 163.7)	163.1 (162.4, 163.7)	0.748
Weight (kg)	63.5 (62.6, 64.5)	63.9 (63.5, 64.4)	63.9 (63, 64.7)	0.736
BMI (kg/m^2^)	23.73 (23.44, 24.01)	23.88 (23.74, 24.02)	23.93 (23.67, 24.18)	0.551
ASA PS (I/II/III)	256 (40)/ 334 (52.2)/ 50 (7.8)	1206 (45.2)/ 1266 (47.5)/ 194 (7.3)	364 (45.5)/ 378 (47.3)/ 58 (7.2)	0.186
Underlying disease (Yes)	324 (50.6)	1311 (49.2)	416 (52)	0.350
Smoking (Yes)	64 (10)	270 (10.1)	81 (10.1)	0.995
Opioid naïve (Yes)	500 (78.1)	2098 (78.7)	651 (81.4)	0.208
Anesthesia duration (h)	2.32 (2.19, 2.45)	2.20 (2.15, 2.26)	2.19 (2.09, 2.28)	0.137
Intraoperative opioid (Yes)	522 (81.6)	2243 (84.1)	676 (84.5)	0.238

The values are expressed as means (95% confidence intervals) or numbers (percentage) of patients. ASA PS, American Society of Anesthesiologists physical status; BMI, body mass index; PCA, patient-controlled analgesia. Opioid naïve: patients without a history of previous opioid intake. Group L, NRS < 4; Group M, 4 ≤ NRS < 7; Group H, NRS ≥ 7 at the 6th postoperative hour.

**Table 2 medicina-57-01065-t002:** Drugs used for intravenous PCA (*n* = 4106).

Variables	Group L (*n* = 640)	Group M (*n* = 2666)	Group H (*n* = 800)	*p*-Value
Opioids				
Fentanyl	627 (98)	2600 (97.5)	774 (96.8)	0.665
Oxycodone	10 (1.6)	49 (1.8)	19 (2.4)	
Sufentanil	3 (0.5)	17 (0.6)	7 (0.9)	
Adjuvant analgesics (yes)	615 (96.1)	2596 (97.4)	768 (96)	0.081
Adjuvant antiemetics (yes)	615 (96.1)	2596 (97.4)	768 (96)	0.062

The values are expressed as numbers (percentage) of patients. PCA, patient-controlled analgesia. Group L, NRS < 4; Group M, 4 ≤ NRS < 7; Group H, NRS ≥ 7 at the 6th postoperative hour.

**Table 3 medicina-57-01065-t003:** Settings and drug doses in intravenous PCA (*n* = 4106).

Variables	Group L (*n* = 640)	Group M (*n* = 2666)	Group H (*n* = 800)	*p*-Value
Settings				
BIR (mL/h)	1.99 (1.98, 2.00)	1.99 (1.98, 1.99)	1.99 (1.99, 2.00)	0.409
Bolus volume (mL/bolus)	1.61 (1.55, 1.66)	1.71 ^†^ (1.68, 1.73)	1.67 (1.62, 1.71)	0.001 *
Lockout interval (min)	22.82 (21.93, 23.71)	24.45 ^†^ (24.08, 24.83)	23.83 (23.08, 24.58)	0.001 *
Doses				
DOSE-FEN-TOTAL (μg) ^§^	1595.20 (1556.76, 1633.65)	1579.67 (1567.85, 1591.50)	1590.19 (1568.07, 1612.30)	0.514
DOSE-FEN-OP (μg)	890.98 (873.49, 908.48)	890.10 (881.93, 898.28)	898.69 (883.49, 913.88)	0.615
DOSE-FEN-NONOP (μg) ^§^	704.22 (672.32, 736.12)	689.57 (683.07, 696.07)	691.50 (678.73, 704.27)	0.343
DOSE-EME (mg)	1.18 (1.18, 1.19)	1.18 ^†^ (1.18, 1.18)	1.17 ^‡^ (1.16, 1.170)	<0.001 *

The values are expressed as means (95% confidence intervals). BIR, background infusion rate; DOSE-EME, dose of antiemetics; PCA, patient-controlled analgesia. *, statistical significance at *p* < 0.05 in one-way ANOVA. ^†^, *p* < 0.05 compared with group L. ^‡^, *p* < 0.05 compared with group M. ^§^, fentanyl-equivalent (μg) doses of opioids (DOSE-FEN-OP), non-opioid adjuvant analgesics (DOSE-FEN-NONOP), and total analgesics (DOSE-FEN-TOTAL) converted using ratios of oxycodone (μg) to fentanyl (100:1), ratios of sufentanil (μg) to fentanyl (1:10), ratio of ketorolac (mg) to fentanyl (25:100), and ratio of nefopam (mg) to fentanyl (20:100). Group L, NRS < 4; Group M, 4 ≤ NRS < 7; Group H, NRS ≥ 7 at the 6th postoperative hours.

**Table 4 medicina-57-01065-t004:** Background infusion rates of opioids, adjuvant analgesics, and adjuvant antiemetics for PCA (*n* = 4106).

Variables	Group L (*n* = 640)	Group M (*n* = 2666)	Group H (*n* = 800)	*p*-Value
BIR-FEN-TOTAL (μg/h) ^‡^	31.78 (31.00, 32.56)	31.45 (31.20, 31.69)	31.71 (31.26, 32.17)	0.439
BIR-FEN-OP (μg/h) ^‡^	17.75 (17.40, 18.11)	17.72 (17.55, 17.89)	17.93 (17.62, 18.24)	0.505
BIR-FEN-NONOP (μg/h) ^‡^	14.03 (13.38, 14.67)	13.73 (13.59, 13.86)	13.79 (13.53, 14.04)	0.337
BIR-EME (μg/h)	23.57 (23.40, 23.74)	23.45 (23.37, 23.53)	23.24 ^†^ (23.08, 23.41)	0.014 *

The values are expressed as means (95% confidence intervals). BIR, background infusion rate; BIR-EME, BIR for adjuvant antiemetics; PCA, patient-controlled analgesia. *, statistical significance at *p* < 0.05 in one-way ANOVA. ^†^, *p* < 0.05 compared with group L. ^‡^, BIRs recalculated as fentanyl-equivalent (μg) doses for opioids (BIR-FEN-OP), non-opioid adjuvant analgesics (BIR-FEN-NONOP), and total analgesics (BIR-FEN-TOTAL) using the ratios of oxycodone (μg) to fentanyl (100:1), ratios of sufentanil (μg) to fentanyl (1:10), ratio of ketorolac (mg) to fentanyl (25:100), and ratio of nefopam (mg) to fentanyl (20:100).

**Table 5 medicina-57-01065-t005:** Requirement for rescue analgesics and antiemetics during intravenous PCA (*n* = 4106).

Variables	Group L (*n* = 640)	Group M (*n* = 2666)	Group H (*n* = 800)	*p*-Value
Rescue analgesic requirement (yes)	106 (16.6)	533 (20)	213 (26.6)	<0.001 *
Rescue antiemetic requirement (yes)	16 (2.5)	67 (2.5)	23 (2.9)	0.843

The values are expressed as numbers (percentage) of patients. PCA, patient-controlled analgesia. *, statistical significance at *p* < 0.05. Group L, NRS < 4; Group M, 4 ≤ NRS < 7; Group H, NRS ≥ 7 at the sixth postoperative hour.

## Data Availability

The data presented in this study are available on request from the corresponding author, through the institutional review board, and reviewers. The data are not publicly available due to restrictions of obtaining approval from the IRB for the disclosure of data. If anyone requires our data for this study, please do not hesitate to contact the corresponding author.

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
