# Peer review of "Cutoff Values for Providing the Ideal Intravenous Patient-Controlled Analgesia According to the Intensity of Postoperative Pain—A Retrospective Observational Study"

_medicina, 2021, doi:10.3390/medicina57101065_

Round 1

Reviewer 1 Report

Major

  • How about performing ROC curve analysis by modeling with all variables such as BIR, bolus, and lockout time for each pain intensity group?
  • Please refer to it.
  • https://bookdown.org/cardiomoon/roc/multiple-2.html
  • The numbers of BIR, bolus volume, lockout time presented in conclusion seems inappropriate to present definitively because their statistical significance varies according to pain intensity.

Minor

  • line 80 & Figure 1, Please check the enrollment period.
  • Figure 1, line 145, Table 3     In the description of Group L,  NRS > 4 has the wrong direction of the inequality symbol.

Author Response

Here is a point-by-point response to the reviewers’ comments and concerns.

Response to Reviewer 1 Comments

  • We used the line number in the manuscript, which maintained the "Track Changes" function in Microsoft Word.

Point 1: How about performing ROC curve analysis by modeling with all variables such as BIR, bolus, and lockout time for each pain intensity group?             Please refer to it. https://bookdown.org/cardiomoon/roc/multiple-2.html

Response 1: Thank you for pointing this out. We performed the ROC curve analysis suggested by the reviewer. As a result of analysis using all variables, AUCs showed 0.656 in the low PPI group, 0.626 in the moderate PPI group, and 0.699 in the high PPI group for recue analgesics requirements. In addition, AUCs showed 0.605 in the low PPI group, 0.579 in the moderate PPI group, and 0.589 in the high PPI group for recue antiemetics requirements.

However, these values were not significantly lower or higher compared with the AUC values for individual variables in this study. Furthermore, our statistical expert suggested that it seemed to be not suitable as an analytic method to obtain the cutoff values of each parameter to configure the ideal PCA regimens, which is the purpose of this study.

Finally, after in-depth discussion with the co-author and statistical expert, we decided not to describe the results and conclusions for further analysis in this manuscript. Please take this into consideration. In addition, the authors revised the title of this manuscript as follows to better reflect the aim of this study.

Lines 2-3: “Cutoff values for providing the ideal intravenous patient-controlled analgesia according to the intensity of postoperative pain–a retrospective observational study”

Point 2: The numbers of BIR, bolus volume, lockout time presented in conclusion seems inappropriate to present definitively because their statistical significance varies according to pain intensity.

Response 2: Thank you for pointing this out. We agree with this comment. As mentioned by the reviewer, the cutoff values calculated in this study showed various statistical significance (p values) depending on the groups classified according to the expected PPI. However, this study was designed to present cutoff values for PCA devices and drug setting for groups classified according to the expected PPI, not to compare the effectiveness of these cutoff values on postoperative analgesia. The authors decided to maintain the presentation of values for the cutoff value, although showing various statistical significance (p values), in that the cutoff values presented in this study can be used as basic data for devices and drug settings for future PCA research. Please consider this point. So, we revised it as shown in the sentence below (in lines 446-451):

“For the optimal or ideal regimens of PCA depending on PPI, we suggest that the adjustment for PCA settings is needed based on a BIR of 1.75 mL/h and bolus volume of 0.5 mL regardless of expected PPI and the lockout interval among PCA setting is needed to adjusted within 12.5 min for cases with a low expected PPI, and within 5 min for those with a moderate or high expected PPI. Therefore, adjustment of the lockout interval should be considered more than those of BIR and bolus volume for the PCA setting.”

Point 3: line 80 & Figure 1, Please check the enrollment period.

Response 3: Thank you for pointing this out. We agree with this comment. We confirmed that the enrollment period described in line 80 was incorrectly typed. So, we revised it as shown in the sentence below (in lines 88-90):

“This study enrolled 4151 patients aged 12–100 years who received intravenous PCA, had an American Society of Anesthesiologists physical status (ASA PS) of I−III, and were scheduled to undergo any elective surgery from January 3, 2019 to December 29, 2020.”

Point 4: Figure 1, line 145, Table 3. In the description of Group L, NRS > 4 has the wrong direction of the inequality symbol.

Response 4: Thank you for pointing this out. We agree with this comment. We found that it was incorrectly typed. So, we revised “NRS > 4” described in all tables and figure legends to “NRS < 4” (in lines 103, 202, 212, 229, 255, 271, and 277).

  • We have revised some words and sentences in the revision process according to comments of reviewers. We have highlighted the changes within the manuscript using the "Track Changes" function in Microsoft Word.

Reviewer 2 Report

Thank you for the article!

Please include in the article the type of surgery and the eventual complications i.e. sedation levels of the patients during the analgesia period. 

As You stated in the title of the article "The ideal regimens..." need an good overview of the type of surgery and complications that you had during the therapy.

Please cite the articles for the conversion of non opioid to opioid analgesics in the 4.3. Limitation of this study paragraph.

Author Response

Here is a point-by-point response to the reviewers’ comments and concerns.

Response to Reviewer 2 Comments

  • We used the line number in the manuscript, which maintained the "Track Changes" function in Microsoft Word.

Point 1: Please include in the article the type of surgery and the eventual complications i.e. sedation levels of the patients during the analgesia period. As you stated in the title of the article "The ideal regimens..." need a good overview of the type of surgery and complications that you had during the therapy.

Response 1: Thank you for pointing this out. We agree with this comment. However, this study was designed to present cutoff values for PCA devices and drug setting for groups classified according to the expected PPI, not to compare the effectiveness of these cutoff values on postoperative analgesia. Moreover, this study included numerous surgeries performed in various departments, making it difficult to describe or analyze all of the surgeries included in the groups classified according to the expected PPI. In addition, the analysis of complications occurring in patients who received PCA does not meet the purpose of this study. The authors think that the description of the analysis results of the types of surgery and complications mentioned by reviewer should be analyzed and described in the RCTs conducted based on the results of this study in the future. Please consider this point. Therefore, the authors revised the title and some sentences of this manuscript as follows to better reflect the aim as well as results of this study.

Lines 2-3: “Cutoff values for providing the ideal intravenous patient-controlled analgesia according to the intensity of postoperative pain–a retrospective observational study”

Lines 11-13: “We investigated the cutoff values for providing the ideal intravenous patient-controlled analgesia (PCA) that could reduce rescue analgesics or antiemetics requirements, based on the grades of postoperative pain intensity (PPI).”

Lines 28-29: “The ideal PCA for reduction of rescue analgesics or antiemetics can be achieved by adjustment of PCA settings and drug dosages carefully with these cutoff values depending on the expected grades of PPI.”

Lines 69-76: “Considering this situation, the development of recommendable cutoff values for PCA settings and drug dosages to provide the ideal intravenous PCA based on clinical situations is essential. However, the evidence for proper fentanyl use in PCA is limited since most studies were conducted with morphine-based regimens [1,7]. Therefore, we performed a retrospective review of electronic medical records to investigate the cut-off values for PCA settings and drug dosages to provide the ideal intravenous fentanyl-based intravenous PCA that can reduce rescue analgesic and antiemetic requirement on the basis of postoperative pain intensity (PPI) regardless of surgical department and surgical type.”

Point 2: Please cite the articles for the conversion of non-opioid to opioid analgesics in the 4.3. Limitation of this study paragraph.

Response 2: Thank you for pointing this out. We agree with this comment. We cite the articles for the conversion of non-opioid to opioid analgesics as shown in the sentence below (in line 427):

 “4.3. Limitations of this study

~. Second, patients receiving various opioids and non-opioid analgesics during PCA were enrolled, and all analgesics were converted into of fentanyl equivalent doses using conversion ratios reported in previous studies [8-11].”

  • We have revised some words and sentences in the revision process according to comments of reviewers. We have highlighted the changes within the manuscript using the "Track Changes" function in Microsoft Word.

Reviewer 3 Report

The authors present a well written and clear manuscript, of unquestionable clinical interest for the daily practice. This complex and time-consuming retrospective study draws attention for an important aspect that “effective PCA can be provided by adjusting the lockout interval and bolus volume rather than BIR, and by applying smaller bolus doses and shorter lockout intervals with an increasing PPI grade.”, among other clinical useful information.

Nevertheless, and as it occurs in several retrospective studies, some variables cannot be controlled, as the authors acknowledge in the discussion. And there are some important variables that were not discussed as limitations of the study that could, in fact, have biased the results. These limitations should also be addressed in the discussion or in completing the methods sections.

Main comments:

Lines 102 to 105- Did these events occur during the intraoperative period or in the six hours postoperative period before the beginning of the study? And what was the shortest time interval between the administration of these antagonistic drugs and the determination of PPI?

Lines 114 -117- How was pain addressed in the patients in the period between the end of the surgery and the assessment of the PPI at the sixth postoperative hour? From what I have understood, BIR was started after surgery. Did the patients need any rescue medication or were free to use the PCA button during the six-hour period prior to the study, while in BIR? If so, did the authors register the time between the last PCA pushed button or the last rescue medication administered in the postoperative period, prior to the assessment of the PPI for allocating the patients to the different groups?

The conclusion in the abstract does not draw the readers’ attention and should be improved. I suggest that authors include lines 338-340 in the abstract conclusion.

Section 3.6 is very confusion. A lot of written data and the graphics (that repeat the written data) disperses the reader’s attention. The data mentioned in the 3.6 subsections should be resumed to tables for better understanding. Graphics are unnecessary.

Another aspect that should be addressed in the discussion, as the authors also mention in lines 48, 49, is the fact that this study was performed in Korean patients and, therefore, caution should be used when extrapolation the results of this study to the general population.

It would also be interesting knowing the patients’ opioid requirements during the surgical intervention, as that can also be an indicator of the distinct surgical noxious stimulation and/or different patients’ pain threshold. Which is an important factor to consider when addressing postoperative pain.

Minor comments:

Lines 145-146- did the authors mean “… along the postoperative 48 hours”? I suggest using the word “analyzed” instead of “investigated”. Also, in several other sentences along the manuscript.

Line 166- I suggest “…repeated”

Line 171- “… of the patients”.

Line 192, 206- please clarify the groups.

Lines 238, 239, 240- “…showed statistical significance.” When compared to what?

Authors should avoid using “we”…

In the discussion, authors should start to indicate the main results of their study. After, authors should discuss the results. When referring to the study mentioned in line 304, authors should provide more detailed information about the results of this study.

Author Response

Here is a point-by-point response to the reviewers’ comments and concerns.

Response to Reviewer 3 Comments

  • We used the line number in the manuscript, which maintained the "Track Changes" function in Microsoft Word.

Main comments:

Point 1: Lines 102 to 105- Did these events occur during the intraoperative period or in the six hours postoperative period before the beginning of the study? And what was the shortest time interval between the administration of these antagonistic drugs and the determination of PPI?

Response 1: Thank you for pointing this out. The adverse events described in lines 102 to 105 means that they have occurred during emergence period as already described in the text. The treatment for them is described in the hospital protocol for intraoperative anesthesia management. We have an obligation to manage intraoperative address events in accordance with this protocol and comply well with them. We described these sentences to show that all patients were managed for anesthesia and adverse events according to the same protocol. Please consider this point.

Point 2: Lines 114 -117- How was pain addressed in the patients in the period between the end of the surgery and the assessment of the PPI at the sixth postoperative hour? From what I have understood, BIR was started after surgery. Did the patients need any rescue medication or were free to use the PCA button during the six-hour period prior to the study, while in BIR? If so, did the authors register the time between the last PCA pushed button or the last rescue medication administered in the postoperative period, prior to the assessment of the PPI for allocating the patients to the different groups?

Response 2: Thank you for pointing this out. Does your comments mean that in evaluating the PPI of postoperative 6 hours, it should be evaluated fairly without bias of data and then analyzed into three groups according to the grade of the PPI? We agree with these comments. However, please consider first that this study was a retrospective observational study, not an RCT study.

Actually, postoperative pain with NRS score > 4 was managed in patients who received postoperative PCA according to the method described in the description in lines 140-146. This was in compliance with the hospital protocol for postoperative pain management and was applied to all patients who received postoperative PCA. So, we allowed all patients pushing demand button and receiving rescue medication during the six-hour period prior to allocation of patients to 3 groups.

As the reviewer mentioned, we also think it may be necessary to analyze the PPI considering the time of the last PCA pushed button or the last rescue medication before allocating patients to one of the three groups according to the PPI at postoperative 6 hours. However, it was impossible to collect data on the time of pushing demand button, the demand counts, the infused counts, the infused opioid doses, and the infused volume, even though they pushed demand button. Their data were not recorded in electronic medical record according to a specified time interval during postoperative period. So, we could not register the time between the last PCA pushed button or the last rescue medication administered in the postoperative period.

In addition, because this study was not designed for specific departments or surgeries, we included all patients who received intravenous PCA regardless of the surgical departments and types of surgery. Furthermore, even if patients undergo the same surgery from a doctor, they show varieties of PPI. Most patients who have undergone surgeries requiring general anesthesia complain of moderate or higher PPI. In general, intravenous PCA is provided for these patients, but in patients who expect low PPI, intravenous PCA is provided by their needs.

Therefore, as you know and we mentioned in limitation of this study, it was difficult to classify the grade of PPI by the surgery types or surgical departments in this retrospective study. Instead, when the data were classified into three groups based on PPI at postoperative 6 hours, the demographic data, including dose and BIR of drugs used in PCA, showed no significant difference. So, we decided to classify data into 3 groups based on PPI at postoperative 6 hours.

Please take these points into consideration.

Point 3: The conclusion in the abstract does not draw the readers’ attention and should be improved. I suggest that authors include lines 338-340 in the abstract conclusion.

Response 3: Thank you for pointing this out. We agree with this comment. The authors revised the abstract conclusion as follow (in lines 30-32).

“Conclusion: The ideal PCA for reduction of rescue analgesics or antiemetics can be achieved by adjustment of PCA settings and drug dosages carefully with these cutoff values depending on the expected grades of PPI. Especially, the ideal PCA can be provided by adjusting the lockout interval and bolus volume rather than BIR, and by applying smaller bolus doses and shorter lockout intervals with an increasing PPI grade.”

Point 4: Section 3.6 is very confusion. A lot of written data and the graphics (that repeat the written data) disperses the reader’s attention. The data mentioned in the 3.6 subsections should be resumed to tables for better understanding. Graphics are unnecessary.

Response 4: Thank you for pointing this out. We agree with this comment. As the reviewer mentioned, the authors also tried to show the data described in section 3.6 in tables. However, for this, six tables were required, which were 6 pages long. So, the authors showed the cutoff values, which are important parts of these tables, in graphics. We also decided them to provide the tables as supplementary tables separately at the end of this manuscript, and we cited them in the text. Please consider this point.

Point 5: Another aspect that should be addressed in the discussion, as the authors also mention in lines 48, 49, is the fact that this study was performed in Korean patients and, therefore, caution should be used when extrapolation the results of this study to the general population.

Response 5: Thank you for pointing this out. We agree with this comment. The authors further described the following sentence at the end of section 4.3 (in lines 422-424).

“Finally, this study was performed in Korean patients. Therefore, caution should be used when extrapolation the results of this study to the general population.”

Point 6: It would also be interesting knowing the patients’ opioid requirements during the surgical intervention, as that can also be an indicator of the distinct surgical noxious stimulation and/or different patients’ pain threshold. Which is an important factor to consider when addressing postoperative pain.

Response 6: Thank you for pointing this out. We agree with this comment. However, please consider first that this study was a retrospective observational study, not an RCT study. Unfortunately, the authors were unable to obtain detailed data on patients' opioid requirements during surgery through medical records.

Minor comments:

Point 7: Lines 145-146- did the authors mean “… along the postoperative 48 hours”? I suggest using the word “analyzed” instead of “investigated”. Also, in several other sentences along the manuscript.

Response 7: Thank you for pointing this out. We agree with this comment. The authors modified the word using "analyzed" instead of "investigated" according to the reviewer's comment (in lines 11-14, 153-170).

“We analyzed the cutoff values for providing the ideal intravenous patient-controlled analgesia (PCA) that could reduce rescue analgesics or antiemetics requirements, based on the grades of postoperative pain intensity (PPI). Materials and Methods: PCA regimens of 4106 patients were retrospectively analyzed, and ~.”

“PCA regimens (types and doses of opioids, adjuvant analgesics, and adjuvant antiemetics) and PCA device settings (BIR, bolus volume, and lockout interval) were analyzed. ~ .

The NRS at the sixth postoperative hour was analyzed to allocate patients into low, moderate, and high PPI groups (group L, group M, and group H, respectively) ac-cording to NRS > 4, 4 ≤ NRS < 7, NRS ≥ 7 [12]. Meanwhile, the use of rescue analgesics and antiemetics was analyzed along the 48th postoperative hour.

~ PONV risk factors (smoking, motion sickness, and previous PONV), anesthesia duration, and intraoperative opioid were also analyzed.”

Point 8: Line 166- I suggest “…repeated”

Response 8: Thank you for pointing this out. The authors checked the sentences again, and concluded that the last sentence was typed incorrectly. So, the last sentence was deleted (in line 189).

“Optimal cutoff values were determined based on the maximum values of the Youden index, calculated by [sensitivity + specificity − 1]. Statistical significance was set at p < 0.05. Then, ROC curve analysis were performed again

Point 9: Line 171- “… of the patients”.

Response 9: Thank you for pointing this out. The authors revised this sentence as follow (in line 194).

“3.1. Characteristics of the patients”

Point 10: Line 192, 206- please clarify the groups.

Response 10: Thank you for pointing this out. Since the definitions of groups were described in the text and tables, the authors decided not to modify them, judging that no additional descriptions were needed for the groups. Please consider this point.

Point 11: Lines 238, 239, 240- “…showed statistical significance.” When compared to what?

Response 11: Thank you for pointing this out. The "statistical signature" described in section 3.6 did not mean comparison between groups, but reported significant p values provided by ROC analysis within that group. Among the results for many variables, it showed what variables showed significant ROC analysis results. Please consider this point.

Point 12: Authors should avoid using “we”…

Response 12: Thank you for pointing this out. Before submitting this manuscript, the authors had received English corrections from experts and submitted the final approved manuscript. At that time, there was no point of using "we" in the expert's opinion, so it is judged that it is not unreasonable to use "we". However, if the reviewer has a problem with the use of "we", the authors will request the matter to be reviewed in the final English correction stage later. Please consider this point.

Point 13: In the discussion, authors should start to indicate the main results of their study. After, authors should discuss the results. When referring to the study mentioned in line 304, authors should provide more detailed information about the results of this study.

Response 13: Thank you for pointing this out. We agree with this comment. The authors fully discussed the reviewer's opinion. The authors decided to describe the last sentence in the first paragraph of the discussion section at the beginning of the discussion section as the main results of this study. In addition, the authors further described the more detailed information about the results of this study cited in the first paragraph of the discussion as follow (in lines 333-341).

“We identified the cutoff values of the settings and drug compositions for the ideal PCA regimen according to the grades of PPI. In general, higher bolus volume, faster BIR, shorter lockout interval, and larger opioid dose in the PCA settings were related to decreased demand for rescue analgesics, while they were risk factors for rescue antiemetic requirement. A previous study analyzed the cutoff values indicating no requirement of rescue analgesics and antiemetics in patients receiving fentanyl-based postoperative PCA [2]. They suggested that a fentanyl BIR should be at least 0.38 μg/kg/h to provide effective postoperative analgesia without administration of rescue analgesics and a fentanyl BIR of over 0.36 μg/kg/h to administer rescue antiemetic.”

  • We have revised some words and sentences in the revision process according to comments of reviewers. We have highlighted the changes within the manuscript using the "Track Changes" function in Microsoft Word.

Round 2

Reviewer 2 Report

I have no more suggestions.

Thank you!

Kind regards

Author Response

Thank you for your kind comments.

if we get the final acceptance for this manuscript, we will upload the final manuscript after receiving the English editing from the native English speaker again.

Reviewer 3 Report

Thank you very much for your response to my suggestions. 

Nevertheless, all the limitations I have mentioned in the first review should be clearly and individually addressed in the manuscript limitations, mentioning in what way these limitations influence the study results and conclusions. The absence of this information in the manuscript is a significant  weakness.

Authors should avoid using personal pronouns in scientific writting. For example "We identified" can be replaced by "In this study it was identified..." and "they" should be replaced by "These authors...", ...

Author Response

Here is a point-by-point response to the reviewers’ comments and concerns.

Response to Reviewer 3 Comments

  • We used the line number in the manuscript, which maintained the "Track Changes" function in Microsoft Word.

Main comments:

Point 1: Nevertheless, all the limitations I have mentioned in the first review should be clearly and individually addressed in the manuscript limitations, mentioning in what way these limitations influence the study results and conclusions. The absence of this information in the manuscript is a significant weakness.

Response 1: Thank you for pointing this out. The authors further described all limitations you have mentioned in the first review (points 1, 2, and 6) according your comments at section 4.3 as follows (in lines 390-432).

“This study had some limitations. First, the AUCs of the cutoff values were relatively low due to the uneven distribution and low incidence of rescue analgesic and antiemetic usage. A randomized controlled trial using data with normal distribution is necessary to support these results.

Second, patients receiving various opioids and non-opioid analgesics during PCA were enrolled, and all analgesics were converted into of fentanyl equivalent doses using conversion ratios reported in previous studies [8-11]. Even though the conversion ratios between opioids are well known and validated, the conversion ratios between opioids and non-opioid analgesics are not.

Third, there were several variables at intraoperative, emergence, and postoperative periods, which could influence the postoperative PPI. Intraoperative opioid consumption during intraoperative period is considered as one of indicators that distinct surgical noxious stimulation or classify the grade of PPI according to surgeries. Some of patients enrolled in this study received naloxone for persistent opioid-related respiratory nonresponse, and flumazenil for persistent sedation during emergence period, in accordance with hospital protocol for perioperative anesthesia management. The postoperative use of demand button for infusion of PCA opioid, rescue analgesics, and rescue antiemetics also could influence the postoperative PPI. Unfortunately, the authors could not register these data because they were not recorded in electronic medical record completely according to a specified time interval during perioperative period. Therefore, it was difficult to classify the grade of PPI by the surgery types or surgical departments in this retrospective study.

Forth, this retrospective study included many surgeries with various PPIs and rates of PONV, but subgroup analysis based on the types of surgery was not performed. Furthermore, the authors did not use the anticipated surgery-specified pain intensity for classification of PPI grades. Actually, even if patients undergo the same surgery from a doctor, they can show varieties of PPI. Therefore, it is very difficult to deter-mine the PPI specifically for each type of surgery [13,14]. In particular, it was very difficult to control risk factors to influence the PPI grade for each surgery, because of the incompleteness of the data as the limitation of retrospective study. However, the data of demography and PCA regimens was shown the minimized bias with no significant difference, after classification of data into three groups based on PPIs at postoperative 6 hours. Therefore, the authors used the PPI grades determined with the actual numeric rate score at the sixth postoperative hour in patients received PCA. To validate these results, a well-designed randomized controlled trial or a retrospective study is necessary to confirm the effective procedure-specific regimens in the future.

Therefore, careful interpretation of the findings of this study is necessary to pro-vide fentanyl-based PCA for effective postoperative analgesia in specified surgeries with different expected PPI, and further research will be required with the dosages and settings presented in this study.

Finally, this study was performed in Korean patients. Therefore, caution should be used when extrapolation the results of this study to the general population. “

Point 2: Authors should avoid using personal pronouns in scientific writting. For example "We identified" can be replaced by "In this study it was identified..." and "they" should be replaced by "These authors...",

Response 2: Thank you for pointing this out. As the reviewer mentioned, the authors revised other suitable words instead of "we" in this manuscript.

Lines 11-13: “Background and Objectives: The cutoff values were analyzed for providing the ideal intravenous patient-controlled analgesia (PCA) that could reduce rescue analgesics or anti-emetics requirements, based on the grades of postoperative pain intensity (PPI).”

Lines 67-71: “Therefore, a retrospective review of electronic medical record was performed to investigate the cutoff values for PCA settings and drug dosages to provide the ideal intra-venous fentanyl-based intravenous PCA that can reduce rescue analgesic and antiemetic requirement on the basis of postoperative pain intensity (PPI) regardless of surgical department and surgical type.”

Lines 309-310: “This study identified the cutoff values of the settings and drug compositions for the ideal PCA regimen according to the grades of PPI.”

Lines 371-375: “These findings suggest that there is no optimal dose and BIR of analgesics for reducing the demand for rescue analgesics and antiemetics, and that it should be considered that if the dose or BIR of PCA drugs is set between these cutoff values to reduce the demand for rescue analgesics and antiemetics, the patients may experience uncontrolled postoperative pain, PONV, or both.”

Lines 382-384: “Thus, it can be set the BIR of antiemetics between 15 μg/h to 25 μg/h considering the risk–benefit ratio between effective analgesia and less adverse events.”

Lines 437-476: “For the optimal or ideal regimens of PCA depending on PPI, the authors suggest that the adjustment for PCA settings is needed based on a BIR of 1.75 mL/h and bolus volume of 0.5 mL regardless of expected PPI and the lockout interval among PCA set-ting is needed to adjusted within 12.5 min for cases with a low expected PPI, and within 5 min for those with a moderate or high expected PPI. Therefore, adjustment of the lockout interval should be considered more than those of BIR and bolus volume for the PCA setting.”

  • We have revised some words and sentences in the revision process according to comments of reviewers. We have highlighted the changes within the manuscript using the "Track Changes" function in Microsoft Word.
